# New Method for the Solidification of High-Concentration Radioactive Borate Solution by Cement-Based Materials

Haosen Ma [1,2], Sensen Yuan [3], Haining Geng [4] and Qiu Li [1,*]

1　State Key Laboratory of Silicate Materials for Architectures, Wuhan University of Technology, Wuhan 430070, China
2　School of Material Science and Engineering, Wuhan University of Technology, Wuhan 430070, China
3　Wuhan Mafangshan-Whut Engineering Structure Co., Ltd., Wuhan 430205, China
4　Hubei Urban Construction Vocational and Technological College, Wuhan 430205, China
*　Correspondence: qiu-li@whut.edu.cn

**Abstract:** Cement is widely used for the solidification of low- and intermediate-level radioactive waste materials. Radioactive borate solution with a high concentration of boron is one of the main radioactive wastes produced in nuclear stations. It is difficult to solidify this solution by using cement because borate has a great inhibitory effect on the cement hydration process. In this study, the hydration kinetics, strength, durability, phase assemblage, and transportation and transformation of the silicon of the paste that blended Portland cement with 5 M borate solution were investigated. After the addition of sodium hydroxide and sodium metasilicate to the paste, the cement hydration process was restarted, and the 28-days strength of samples met the requirements of the Chinese standard. The mechanism of overcoming the retardation of cement hydration by the borate solution was attributed to the formation of calcium metaborate, ettringite, portlandite, and calcium silicate hydrate with the restarting of cement hydration, without the formation of ulexite.

**Keywords:** borate solution; Portland cement; hydration products; calcium metaborate; solidification

## 1. Introduction

A pressurized water reactor (PWR) is a type of nuclear power reactor [1]. The primary system of PWRs [2] contain high concentrations of radioactive borate solution. Boron is used as a strong thermal-neutron absorber [3–5] together with water to decelerate and absorb the neutrons produced by the reactor. The waste effluent that contains a high-concentration borate solution is generated during the replacement of the water circulating in the primary system, and it is usually classified as a low-level waste (LLW) [6,7]. A large amount [8] of LLW has to be immobilized properly to protect the environment and humans [9].

Cement is the commonly used material for the solidification of LLW. However, borate in the waste solution has a strong retarding effect on the hydration process of cement mainly because a coating layer of borate [10,11] covers the surface of the cement clinker and inhibits cement hydration. The strong hydration inhabitation leads to a remarkable extension of the cement-setting time [12], lower compressive strength [13], and lower durability because of which the samples do not meet the performance requirements.

The hydration retardation product of borate solution with cement has been well investigated: it iscomposed of amorphous or poorly crystallized calcium borate hydrates [10,11]. Calcium sulphoaluminate (CSA) cement was immersed in 0.5 M borate solution to investigate the formation and phase assemblage of the retardation product [8]. The results indicated the formation of a coating layer composed of crystallized ulexite ($NaCaB_5O_9 \cdot 8H_2O$) which inhibited the dissolution and hydration process of the CSA clinker. The ulexite layer could maintain stability when the pH value is lower than 10.8 and became unstable and dissolved when the pH value is higher than 12.5 [14]. Thus, the crucial point of

high-concentration borate solution solidification by cement is to overcome the hydration retardation by the increase in pH in cement with borate solution.

Thus far, several ways were proposed to solve this problem. One solution is the addition of NaOH—alite hydration can be accelerated by NaOH [15,16] and ulexite [10] is transformed into other products so that the hydration can be restarted. In addition, portlandite [17,18] can overcome the inhibitory effect of borate and form calcium hexahydroborite with Portland cement. However, the unstable calcium hexahydroborite could transform into calcium mono- or quadri-boroaluminate, which causes the expansion of cement paste and reduction in strength and durability [19]. LiOH [20] was used to overcome the cement paste inhibition caused by the borate solution and to accelerate dissolution of the covering layer on the C-S-H nuclei, and the retardation effect caused by borated was counteracted. In addition, NaOH and sodium aluminate [16] were used for the immobilization of high concentration borate solution with Portland cement. The results indicated that with the addition of NaOH and sodium aluminate in borate solution, the ulexite layer, which covered the clinker, dissolved, and the hydration process was restarted with the formation of boron-contained AFt and gowerite.

Similarly, CSA cement can solidify radioactive borate effluent. Simulated borate effluent and borate resins can be immobilized by CSA cement blended with calcium hydroxide ($Ca(OH)_2$) and lithium hydroxide (LiOH) [21]. LiOH [19] can accelerate the formation of Li-containing aluminate hydroxide in CSA cement. The addition of LiOH was beneficial to the fast precipitation of Li-containing aluminum hydroxide, which can accelerate the hydration process of CSA. The acceleration caused by LiOH counteracted the retardation by sodium borate, so that the hydration process of CSA started. CSA cement blended with Portland cement can be used to solidify the borate solution [22]. The maximum salinity reached 600 g/L including borates can be immobilized in this material in that mixture at room temperature. The boron can be encapsulated in structure of ettringite and calcium monosulfoaluminate so that the borate concentration in pore solution decreased and the hydration restarted.

Another solution is to use alkali-activated cementitious materials for immobilizing borate solution. Slag cement, which is activated by sodium metasilicate [23] was used to solidify a borate solution, with a borate concentration of up to 200 g/L. Alkali-activated fly ash cement [24] was used to immobilize the 1.4 M boric acid solution. The occurrence of boric acid did not retard the hydration process, hardening process, and compressive strength of alkali-activated fly ash cement. Additionally, the leachability of alkali-activated fly ash cement was better than Portland cement.

The aim of this study was to investigate the effect of sodium metasilicate with NaOH on the hydration process of cement waste. The hydration kinetics, strength, durability, phase assemblage, and $^{29}$Si nuclear magnetic resonance (NMR) spectra were analyzed and discussed by adopting a series of analytical techniques. The solidified borate samples met the requirements of the Chinese standard.

## 2. Materials and Methods

### 2.1. Materials

The raw materials used in the cement paste were ordinary Portland cement (OPC), sodium hydroxide (NaOH), sodium metasilicate ($Na_2SiO_3 \cdot 9H_2O$), boric acid ($H_3BO_3$), and deionized water. OPC (P.I. 52.5) was provided by Huaxin Cement Co., Ltd. (Wuhan, China). The chemical composition of OPC is listed in Table 1. NaOH (AR), $Na_2SiO_3 \cdot 9H_2O$ (AR), and $H_3BO_3$ (AR) were provided by Sinopharm.

**Table 1.** Chemical composition of OPC.

| Oxide | $SiO_2$ | $Al_2O_3$ | $Fe_2O_3$ | CaO | MgO | $SO_3$ | $Na_2O$ | $P_2O_5$ | LOI |
|---|---|---|---|---|---|---|---|---|---|
| wt% | 19.35 | 4.57 | 3.75 | 61.21 | 2.65 | 3.64 | 0.20 | 0.13 | 3.09 |

### 2.2. Experiments

The borate solution concentration was 5 M. To prepare the solution, first, NaOH is added to water, and then $H_3BO_3$ is added to the solution. Table 2 lists the composition of the borate solution.

**Table 2.** Mix design of high-concentration borate solution.

| Component | $H_3BO_3$ | NaOH | Water |
|---|---|---|---|
| wt(%) | 22.72 | 3.81 | 73.47 |

Table 3 lists the mix design of the cemented form of borate solution. The mass content of $Na_2O$ in $Na_2SiO_3$ to cement is 0% to 5% of the cement. The samples were blended with OPC, borate solution, NaOH, and $Na_2SiO_3$ with a solution/solid ($s/s$) ratio of 0.6. These pastes were mixed for 3 min. Subsequently, the grouts were put into Φ50 mm × 50 mm cylinder molds, and the surface was covered with a plastic wrap for one day. Then, the samples were demolded and cured in sealed bags for the compressive strength test and durability tests.

**Table 3.** Mix design of cemented waste form.

| | OPC (g) | NaOH (g) | $Na_2SiO_3 \cdot 9H_2O$ (g) | Borate Solution (g) |
|---|---|---|---|---|
| S0 | | | 0 | |
| S1 | | | 4.58 | |
| S2 | | | 9.16 | |
| S3 | 100 | 5 | 13.75 | 60 |
| S4 | | | 18.32 | |
| S5 | | | 22.92 | |

### 2.3. Analysis Techniques

For calorimetry (TAM Air, TA Instruments, New Castle, DE, USA), samples were mixed by hand for approximately 2 min and then introduced into the calorimeter vessel. For the tests, 16 g of each sample with 7 g of water for reference were used, and the collecting time for calorimetry was 7 days. The detection temperature was 25 °C.

The compressive strength of the samples was tested by using a WAY-300N electro-hydraulic press with a loading rate of 0.6 kN/s. The samples of each mix at 3 days and 28 days were used to test the compressive strength. Three samples were used in each batch. The compressive strength was calculated from the average of three samples. The 28-day strength of the samples should be greater than 7 MPa according to the Chinese standard GB

Freeze–thaw tests were conducted according to th Chinese standard GB 14569.1-2011 [25]. The samples that met the 28-day strength requirement as per this standard were sealed in bags and placed in a cold environment at a temperature of −20 °C for 4 h; thereafter, they were placed in a water bath at 25 °C for 4 h. The freeze–thaw cycles were performed five times, and subsequently, the compressive strength of the samples was tested by applying a loading rate of 0.6 kN/s. Three samples were used for each batch.

The samples that were used for phase assemblage analysis and microstructure analysis were cured in sealed centrifuge tubes at room temperature for 3, 7, 14, or 28 days. At the chosen time, the samples were cut into 5-mm-thick slices and immersed in isopropanol for 8 h, dried for 8 h in a dryer at 40 °C, and finally, placed in sealed bags for the experiments.

X-ray diffraction (XRD) spectra were measured by using a RigaKu MiniFlex600 desktop X-ray diffractometer. The instrument settings were as follows: a scanning range of 3° to 67° with a rate of 5°/min in steps of 0.02° at 40 kV voltage and 15 mA current. For this measurement, the samples had to be ground in an agate mortar to achieve fineness less than 75 µm. The XRD data were analyzed by PANalytical HighScore Plus with PDF2004 database.

Thermogravimetric–derivative thermogravimetry (TG-DTG) analysis data were collected by using a Netzsch STA 499 F3 thermal analyzer over a temperature range of 30–980 °C and at a heating rate of 10 °C/min in nitrogen atmosphere.

[29]Si magic angle spinning (MAS) nuclear magnetic resonance (NMR) spectra were acquired on an Agilent 600M spectrometer. Single pulse [29]Si MAS NMR tests were performed with a 7mm probe at a spinning speed of 10 kHz, relaxation delay of 10 s, and scan number of 365. The specific deconvolutions were according to the method of Yan et al. [26]. The mean silicate chain length (MCL) was calculated using Formula (1) and the Al/Si molar ratio was calculated using Formula (2).

$$MCL = \frac{2\left[Q^1 + Q^2 + \frac{3}{2}Q^2(1Al)\right]}{Q^1} \tag{1}$$

$$Al/Si_{gel} = \frac{\frac{1}{2}Q^2(1Al)}{Q^1 + Q^2 + Q^2(1Al)} \tag{2}$$

## 3. Results and Discussion

### 3.1. Hydration Kinetics

Figure 1a,b show the calorimetric results of mixtures S0–S5. The results show that borate solution exerts a strong retardation effect on cement hydration, in agreement with the existing research. With the addition of only 0% to 1% alkali content of $Na_2SiO_3$, the total heat of S0 and S1 still did not overcome the inhibitory effect of the borate solution mainly because of the low alkalinity in S0 and S1 paste [16]. When a sufficient amount of $Na_2SiO_3 \cdot 9H_2O$ was added, the hydration process that stopped because of the borate solution was restarted. With incremental addition of $Na_2SiO_3$, the heat flow peaks appeared early, but the total heat evolved increased and then decreased. S2 sample had the highest total heat of 160 kJ/g among these samples. However, S2 sample still reached the final setting time within the first two days. Based on these results, samples S3, S4, and S5 were selected for the compressive strength test.

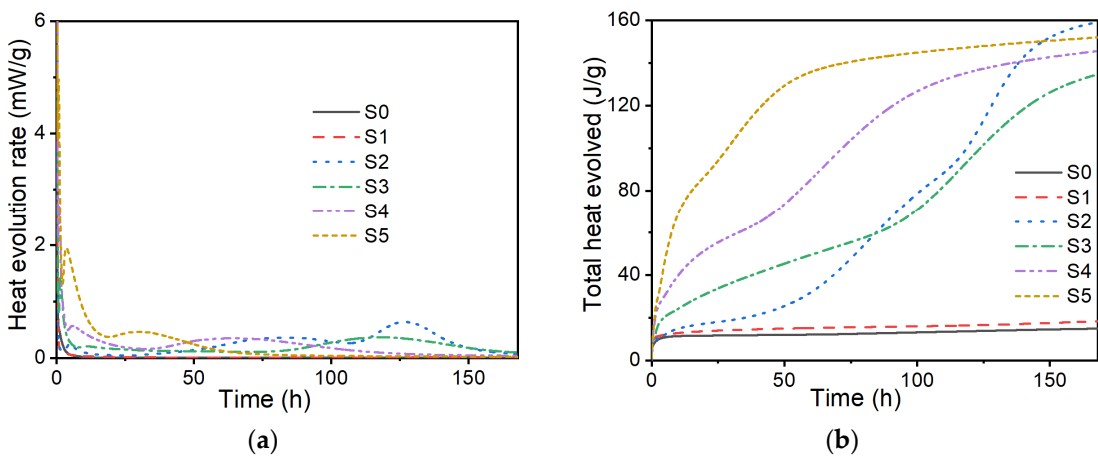

(a)　　　　　　　　　　　(b)

**Figure 1.** (**a**) Heat evolution rate and (**b**) total heat evolved from the mixtures during the first 168 h of hydration.

### 3.2. Strength and Durability

The compressive strength and durability of S3, S4, and S5 samples are shown in Figure 2 and Table 4, respectively, and the setting time of the S2 sample was too long to conduct the compressive strength test. Mixes of S0 and S1 did not set after 28 days of curing. As a result, only mixes of S3, S4, and S5 were tested on their compressive strength and performed other characterizations. The results indicates that the compressive strength of samples increased with curing ages and that the 28-day strength of all the samples was higher than 7 MPa, which is the limit given in the Chinese standard. In addition,

the strength loss of grouts after the freeze–thaw test was less than 25%, which meets the requirements of the Chinese standards. These results show that the hydration process of samples with sufficient addition of NaOH and $Na_2SiO_3$ can overcome the inhibitory effect of the borate solution, allowing the hydration process to restart, and the samples also meet the requirements of the Chinese standards.

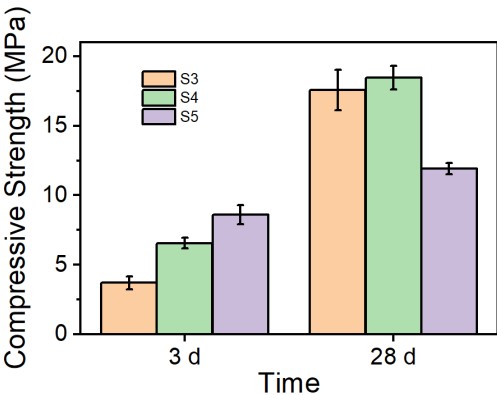

**Figure 2.** Compressive strength with S3, S4, and S5 samples hydrated 3 days and 28 days.

**Table 4.** Compressive strength and durability of samples.

| Group | 3 d Compressive Strength (MPa) | 28 d Compressive Strength (MPa) | Freeze-Thaw Test (MPa) | The Loss of Strength |
|---|---|---|---|---|
| S3 | 3.7 | 17.6 | 15.7 | 10.8% |
| S4 | 6.6 | 18.5 | 14.3 | 22.7% |
| S5 | 8.6 | 11.9 | 14.3 | −20.2% |

*3.3. Phase Assemblage*

The crystalline mineral phase assemblages of samples S3–S5 at different ages is shown in Figure 3. The results indicate the hydration products were all composed of portlandite [27,28], ettringite [29], and calcium metaborate ($CaB_2O_4\cdot2H_2O$). With the increase in $Na_2SiO_3$ content, the intensity of calcium metaborate at 28 days increased and then decreased, showing the same trend as the compressive strength results at 28 days. In these samples, the intensity of calcium metaborate increased and that of alite decreased with curing age. With the addition of a sufficient amount of sodium hydroxide and sodium metasilicate, the ulexite retardation layer that covered the surface of the cement clinker dissolved and decomposed [30] and the hydration process restarted, showing the same tendency as the calorimetry results. Nevertheless, the residues of anhydrate cement still existed, and therefore, the compressive strength of the paste was lower than that of cement blended with same amount of water.

The thermal analysis of samples S3–S5 at different ages is shown in Figure 4. All the TG curves of the samples that can be activated have three regions, namely the 30–300 °C region, 400–500 °C region, and 600–800 °C region. In the first region, the dehydration process of several hydration products occurred. C-(A)-S-H gel [31], ettringite [32,33], and other hydrates lost the interlayer water as $CaB_2O_4\cdot2H_2O$ was dehydrated at 131 °C [28]. The dehydroxylation of portlandite [34] occurred in the 400 °C–500 °C range. The 600–800 °C region corresponds to the decarbonation of calcium carbonate [35]. Quantitative thermal analysis of the weight loss in samples S3–S5 at different ages are listed in Table 5. The results indicate that the weight loss of portlandite and calcite increased at 28 days with the addition of sodium metasilicate; the results indicate that the hydration process of S4 is the highest among these three samples. Further, with the addition of $Na_2SiO_3$, the amount of chemically bound water and calcite increased.

**Figure 3.** X-ray diffraction (XRD) patterns at different ages for samples (**a**) S3, (**b**) S4, and (**c**) S5, and (**d**) those of different samples at 28 days.

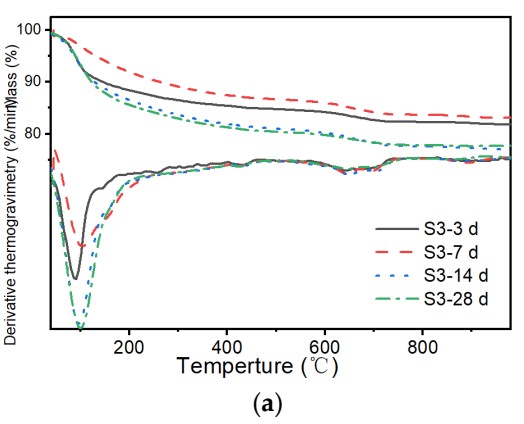

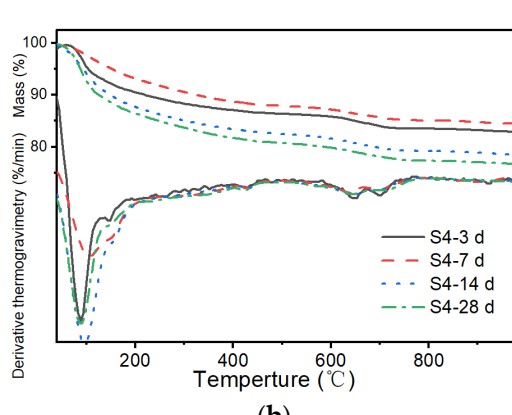

**Figure 4.** *Cont.*

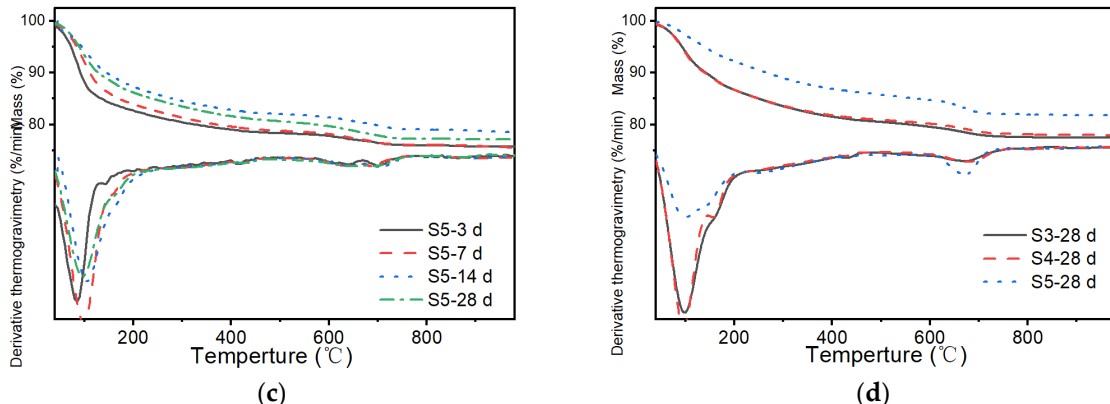

**Figure 4.** Thermal analysis at different ages for (**a**) S3, (**b**) S4, and (**c**) S5, and (**d**) thermal analysis of different samples at 28 days.

**Table 5.** Quantitative thermal analysis of the weight loss in samples S3–S5 at different ages (wt%).

| Temperature | 30–300 °C | 400–500 °C | 600–800 °C |
|---|---|---|---|
| | Chemical bound water and free water | Portlandite | Calcite |
| S3-3d | 13.60 | 0.62 | 1.90 |
| S3-7d | 10.92 | 0.78 | 2.26 |
| S3-14d | 15.64 | 0.89 | 2.60 |
| S3-28d | 16.74 | 0.85 | 1.94 |
| S4-3d | 10.45 | 0.72 | 2.32 |
| S4-7d | 9.48 | 0.82 | 2.08 |
| S4-14d | 14.70 | 0.90 | 2.38 |
| S4-28d | 16.23 | 0.97 | 2.49 |
| S5-3d | 19.49 | 0.71 | 1.80 |
| S5-7d | 18.52 | 0.82 | 2.05 |
| S5-14d | 15.47 | 0.83 | 2.36 |
| S5-28d | 16.38 | 0.98 | 2.41 |

### 3.4. Transportation and Transformation of Silicon

Figure 5 shows the $^{29}$Si MAS NMR spectra of S4 after 3 and 28 days. Four main different environments of silica in the paste were detected through $^{29}$Si MAS NMR. The chemical shift of anhydrate cement ($Q^0$) was 71.3 ppm, while $Q^1$ was −79 ppm [36], $Q^2$(1Al) was −81.5 ppm [37], and $Q^2$ was −83.5 ppm [38]. Table 6 lists the relative fractions, Al/Si ratio, and MCL as obtained by deconvolution of $^{29}$Si MAS NMR spectra.

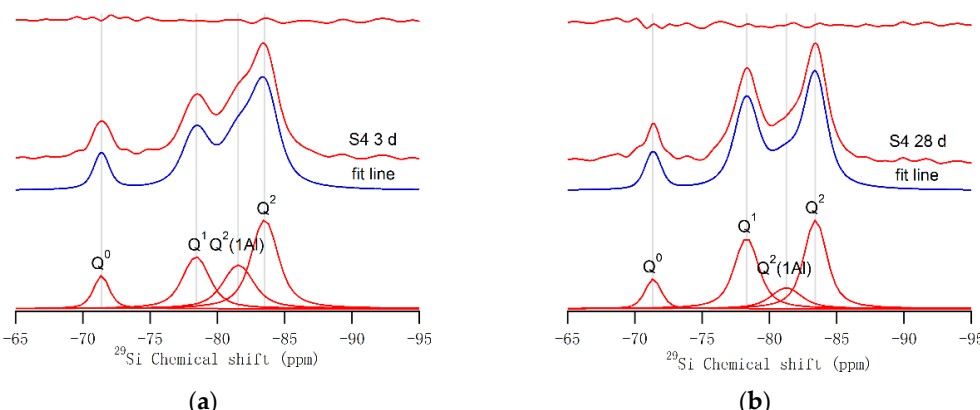

**Figure 5.** $^{29}$Si magic angle spinning (MAS) nuclear magnetic resonance (NMR) spectra of (**a**) S4 after 3 days and (**b**) S4 after 28 days.

**Table 6.** Relative fractions, Al/Si ratio, and mean silicate chain length (MCL) obtained by the deconvolution of $^{29}$Si MAS NMR spectra.

| Sample | $Q^0$ | $Q^1$ | $Q^2$(1Al) | $Q^2$ | Al/Si Molor Ratio | Hydration Degree | MCL |
|---|---|---|---|---|---|---|---|
| S4 3 d | 10.89% | 24.23% | 23.37% | 41.51% | 13.11% | 89.11% | 8.32 |
| S4 28 d | 10.56% | 35.04% | 13.26% | 41.14% | 7.41% | 89.44% | 5.48 |

With the increase in curing age, the frequency of $Q^1$ increased, while the intensity of $Q^2$(1Al) decreased, indicating that the mean chain length was shortened; the same trend was observed in the deconvolution results, which is in agreement with the existing works [26,39,40]. The low of Al/Si ratio indicates that more silicon originating from $Na_2SiO_3$ enters the silica chain. the shorter MCL indicates that the Ca/Si ratio of calcium silicate hydrate (C-(A)-S-H) gels increased. Further, the hydration degree of the cement paste increased with the curing age. The chemical shift of $Q^1$, $Q^2$(1Al), and $Q^2$ was caused by the addition of NaOH and $Na_2SiO_3$.

## 4. Discussion

According to the results from this study, the main hydration products of Portland cement, NaOH, and $Na_2SiO_3$ with 5 M high-concentration borate solution, were calcium metaborate and portlandite. L. J. Csetenyi et al. [10] found that ulexite formed in the system of $Na_2O$-CaO-$B_2O_3$-$H_2O$ without, and ulexite transformed into other products containing boron with the addition of CaO or $Na_2O$. Jean-Baptiste Champenois et al. [14] found that the poorly crystallized borate compound (ulexite) formed on the early hydration process in the paste blended with CSA cements and borate solution. The results indicated that ulexite keeps stable with the pore solution pH of 10.8. Moreover, with the addition of gypsum, the pore solution pH increases to 13 rapidly, and the ulexite becomes destabilized and dissolved. According to the result of Céline Cau Dit Coumes et al. [19], the retardation effect caused by ulexite has been detected and in the first 40 h ulexite was found through thermal analysis and the X-ray diffraction peak of ulexite was detected after 90 days hydration. In addition, the ulexite layer was observed directly by Wei Chen et al. [8], the 100nm thick foil-like ulexite covered the surface of CSA clinker and prevented the hydration process till 28 days. According to the previous studies [16], when pH value of the solution is lower than 12, during the early age hydration of Portland cement with high concentration of borate solution, a dense layer of ulexite formed on the surface of cement clinker and prevented the dissolution of clinker.

Therefore, without the addition of $Na_2SiO_3 \cdot 9H_2O$ or with addition of 1 wt% alkali content of $Na_2SiO_3 \cdot 9H_2O$, the pH value of pore solution in S0 and S1 samples was not high enough to prevent ulexite formation, resulting in the ulexite covering the cement clinker and preventing the dissolution of alite to release the calcium ions into solution, and the hydration process was inhibited with no obvious heat evolution generated in calorimetry test. With the addition of NaOH and more than 1 wt% alkali content of $Na_2SiO_3 \cdot 9H_2O$, the pH value of pore solution of cement paste increased, and the dense ulexite layer became unstable and dissolved subsequently, resulting in the restart of the hydration process of Portland cement and hardening process started, the continuous increment of compressive strength with curing time and the hydration products portlandite, ettringite, calcium metaborate, and C-(A)-S-H gels generated.

After the hydration process started, the C-(A)-S-H gels, portlandite, ettringite, and calcium metaborate formed and the hardening process started, resulting in strength development. After 3 days of hydration, calcium metaborate formed. Additionally, because of the increase in pH value, boron existed as $B(OH)_4^-$ and entered the structure of ettringite, and formed boron-contained ettringite [12,14,41]; however, the intensity of ettringite was too low to distinguish the specific position of peak of boron-containing ettringite in this study. The MCLs of C-(A)-S-H gels in the 3 d and 28 d specimens with $Na_2SiO_3 \cdot 9H_2O$

addition were higher than those in the OPC mixed with water, which is usually around 3.7 [42], indicating a higher degree of polymerization of C-(A)-S-H with the addition of sodium silicate. The reason is that the addition of $Na_2SiO_3 \cdot 9H_2O$ provided extra silicon tetrahedra during the hydration, the proportion of $Q^2$ increased and resulted in a longer MCL [43]. Moreover, the MCL of C-(A)-S-H gels in the specimen with 4 wt% $Na_2SiO_3 \cdot 9H_2O$ addition at 3 days of hydration was higher than that at 28 days of hydration, possibly due to the addition of $Na_2SiO_3 \cdot 9H_2O$, which needs to be explored further. Furthermore, the Al/Si molar ratio of samples at 3 days was higher than that at 28 days, and the Al/Si molar ratio of samples at 3 days is higher than 10% (a common Al/Si molar ratio in C-(A)-S-H of Portland cement with the addition of $Na_2SiO_3 \cdot 9H_2O$), which is an interesting point for the future research. The compressive strength of S3, S4 and S5 samples at 28 days were higher than 7 MPa, which satisfied the requirement of Chinese standard GB 14569.1-2011.

## 5. Conclusions

The main conclusions can be drawn as follows:

1.  A new method for the solidification of high-concentration borate solution by cement-based materials was devised, and it was found that the addition of a sufficient amount of sodium hydroxide and sodium metasilicate can help overcome the retardation effect of borate and restart the cement hydration process.
2.  The 28-day compressive strength of samples S3–S5 samples was higher than 7 MPa, and the strength loss after freeze–thaw tests was less than 25% percentage. The hydration products of cement paste were portlandite, ettringite, calcium metaborate, and C-(A)-S-H gels. With prolonged curing time, the degree of hydration of the cement pastes increased, and the Al-to-Si ratio and MCL decreased.

**Author Contributions:** Conceptualization, Q.L. and H.M.; methodology, H.G. and H.M.; software, H.M.; validation, Q.L., H.G. and S.Y.; formal analysis, H.M. and H.G.; investigation, H.M.; resources, Q.L.; data curation, H.M.; writing—original draft preparation, H.M.; writing—review and editing, Q.L., S.Y. and H.G.; visualization, H.M.; supervision, Q.L.; project administration, Q.L.; funding acquisition, Q.L. All authors have read and agreed to the published version of the manuscript.

**Funding:** This research was funded by the National Natural Science Foundation of China (project 52072279), the State Key Laboratory of Silicate Materials for Architectures (Wuhan University of Technology) (SYSJJ2021-12) and Shenzhen Science and Technology Plan Collaborative Innovation Project (CJGJZD202006617102601003).

**Institutional Review Board Statement:** Not applicable.

**Informed Consent Statement:** Not applicable.

**Data Availability Statement:** Data are available on request.

**Conflicts of Interest:** The authors declare no conflict of interest.

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
