# Peer review of "New Method for the Solidification of High-Concentration Radioactive Borate Solution by Cement-Based Materials"

_jcs, doi:10.3390/jcs6120392_

Round 1

Reviewer 1 Report

Dear Authors

I find the article useful and valuable, however, I would like to ask you to take into account the clique of comments and correct the mistakes.

1. Please correct the chemical subscripts: lines 49, 53,61, 63,64,

2. and also 29Si NMR – „29”  superscript

3. Line 71 - remove bold from "table 3"

4. Subsection 2.3 should be on the next page

5. In Figure 3 you can practically not see the peaks responsible for the dehydroxylation of portlandite and the decomposition of calcium carbonate. Perhaps the scale should be increased.

6. I believe the study should include a reference sample - which is if I understand S0 correctly

7. Subsection 4 - should be "Conclusions"

Regards

Reviewer

Reviewer 2 Report

The presented results are interesting and gave information on materials behaviour. The main thing I am missing in the submitted paper is more complex analysis of functional properties of the developed materials as only tests of the bulk density, and porosity, which mainly influence the material behaviour. Therefore, following suggestions and recommendations could be considered in the revised version of the paper.

1) I miss information of measuring uncertainty of performed analysis.  It must be completed.

2) Please, correct the chemical formulas, lines 49, 53, 61, 63, 64 and 29Si line 55.

3) Were the measurements repeated to get good reproducibility of results?

4) Which devices were specifically used for measurement?

5) Did you try leaching test to verify immobilization of radioactive boron?

Round 2

Reviewer 1 Report

Dear All

Hereby I recommend the article for publication.

Reviewer 2 Report

Thank you for a good job and a well-written paper.

Reviewer 3 Report

Thank you to the authors for making corrections to the article. I believe that the article thus revised is suitable for publication.